# Nutrition-Related N-of-1 Studies Warrant Further Research to Provide Evidence for Dietitians to Practice Personalized (Precision) Medical Nutrition Therapy: A Systematic Review

**DOI:** 10.3390/nu15071756

**Published:** 2023-04-04

**Authors:** Margaret Allman-Farinelli, Brianna Boljevac, Tiffany Vuong, Eric Hekler

**Affiliations:** 1Discipline of Nutrition and Dietetics, Susan Wakil School of Nursing and Midwifery, Faculty of Medicine and Health, The University of Sydney, Camperdown, NSW 2006, Australia; 2The Charles Perkins Centre, The University of Sydney, Camperdown, NSW 2006, Australia; 3The Design Lab, University of California San Diego, San Diego, CA 92093, USA; 4Herbert Wertheim School of Public Health and Human Longevity Science, University of California San Diego, San Diego, CA 92093, USA

**Keywords:** N-of-1 study, diet, medical nutrition therapy, personalized nutrition, precision nutrition, genomic nutrition

## Abstract

N-of-1 trials provide a higher level of evidence than randomized controlled trials for determining which treatment works best for an individual, and the design readily accommodates testing of personalized nutrition. The aim of this systematic review was to synthesize nutrition-related studies using an N-of-1 design. The inclusion criterion was adult participants; the intervention/exposure was any nutrient, food, beverage, or dietary pattern; the comparators were baseline values, a control condition untreated or placebo, or an alternate treatment, alongside any outcomes such as changes in diet, body weight, biochemical outcomes, symptoms, quality of life, or a disease outcome resulting from differences in nutritional conditions. The information sources used were Medline, Embase, Scopus, Cochrane Central, and PsychInfo. The quality of study reporting was assessed using the Consort Extension for N-of-1 trials (CENT) statement or the STrengthening Reporting of OBservational Studies in Epidemiology (STROBE) guidelines, as appropriate. From 211 articles screened, a total of 7 studies were included and were conducted in 5 countries with a total of 83 participants. The conditions studied included prediabetes, diabetes, irritable bowel syndrome, weight management, and investigation of the effect of diet in healthy people. The quality of reporting was mostly adequate, and dietary assessment quality varied from poor to good. The evidence base is small, but served to illustrate the main characteristics of N-of-1 study designs and considerations for moving research forward in the era of personalized medical nutrition therapy.

## 1. Introduction

The randomized controlled trial (RCT) has been a mainstay of evidence-based practice in medicine for many decades [1]. Findings from meta-analysis of two or more RCTs are considered the highest level of evidence when testing the efficacy or effectiveness of a treatment or preventive program [2,3]. However, within an RCT, participants usually demonstrate a wide variety of responses, indicating that although the mean effect may be significant, it will be less or not at all effective for some individuals [4]. This finding similarly applies to evidence-based medical nutrition therapy [5,6]. RCTs typically involve measurement at the baseline and endpoint of a study, with a comparison of endpoints or change from baseline to endpoint between groups. However, multiple measurements of individuals at many time points reveal that the individual variance may be two-to-four times that of the group [4]. Additionally, it has been demonstrated that meta-analyses of RCTs of behavioral interventions (of which nutrition therapy is one) may be misleading about progress in therapy, because interventions may show a positive correlation of effect size with time, and changes in comparators may also influence the outcomes [7]. Pooled effect sizes will converge on a central tendency over time, and confidence intervals diminish. Thus, it is difficult to show increases in efficacy over time using cumulative meta-analyses. Heterogeneity is a major factor in meta-analyses of behavioral interventions because the interventions themselves are heterogenous, complex, multicomponent, and delivered with varying fidelity. Time, context and individual biology will contribute to whether the desired outcome is achieved, and the controls used to supposedly account for these factors demonstrate considerable heterogeneity, with changes between RCT interventions; for example, usual care as a control may improve [7]. Therefore, interpretation of effect sizes from meta-analyses of medical nutrition therapies can be flawed by heterogeneity [7]. 

Precision medicine recognizes that individual responses differ due to a variety of factors such as genetics, lifestyle, and environment. While initially, precision medicine may have been perceived as focusing on molecular biology, the broader view recognizes non-biological determinants to be of equal or even greater significance [8]. When a lack of efficacy is found for dietary treatments, in addition to the level of compliance with the prescribed diet plan, the influences of genome, microbiome, social, psychological, educational, economic, and physical environment must be considered.

In 1986, Guyatt et al. highlighted a study design to account for individual patient variations when trying to discern the best treatment plan for an individual: the N-of-1 randomized trial [9]. This study design uses single patients as their own control and usually involves testing several interventions administered in a randomized fashion with wash-outs in between; it also includes many repeated measurements [10]. Typically, an N-of-1 trial in its simplest form includes a treatment, A, and another treatment or control, B. Quasi-experimental designs may be AB, and a randomized trial AB BA. Additional cycles using the treatments will be carried out until sufficient cycles have been conducted to make a clinical decision. It is also possible for interventions with three or more arms to be tested [11,12]. These trials provide a higher level of evidence for making decisions for individual patients than an RCT [13,14,15]. Additionally, they readily accommodate patients with numerous co-morbidities and care requirements who would usually be excluded from RCTs.

In the behavioral sciences, N-of-1 studies are inclusive of intervention trials and observational designs in which individuals are followed over extended periods of time with no intervention [16]. This allows the relationships between behavior and its predictors to be studied over long durations, and can be used to test and formulate theories of behaviors as well as to refine the final personalized therapy for an individual [17]. Eating is a behavior, so both observational and randomized trial N-of-1 designs would be useful in the study of personalized medical nutrition therapy and preventive nutrition [18]. N-of-1 trials can prioritize disease status, individual characteristics, and circumstances that affect health and adoption of nutrition advice, and can also consider the personal preferences of the individual involved by involving personalized dietary management. This means personalized treatment programs that can adapt to the individual reactions of the person will optimize the final selection of the best dietary management for them. Dietitians routinely personalize their diet plans, education, and counselling when providing medical nutrition therapy [19], but have not routinely had access to data from the genome, microbiome, or real-time tracking of behaviors in different contexts. Developing the evidence base for the practice of medical nutrition therapy and preventive nutrition using N-of-1 studies may offer fruitful research possibilities.

The primary aim of this systematic review was to identify and synthesize the literature on all nutrition-related N-of-1 studies. This review asked the following questions: What types of nutrition interventions have been researched? What medical and public health conditions have been studied? What methods have been employed for dietary assessment? The additional aims were to assess the quality of the overall reporting of the studies and of the dietary assessment methods, both being important to ascertain the level of certainty around studies’ conduct and findings.

## 2. Methods

The protocol for this systematic review has been registered in the Open Science Framework (https://osf.io/nc8em) (registered on 16 July 2022). This review was conducted in accordance with the Preferred Reporting Items for Systematic reviews and Meta-analyses (PRISMA) 2020 guidelines [20], which is recommended in Enhancing the QUAlity and Transparency Of health Research, EQUATOR (https://www.equator-network.org/) (accessed on 28 February 2022).

### 2.1. Eligibility Criteria

The inclusion criteria for this review were studies with (1) an N-of-1 study design; (2) an adult population (aged 18 years or older); (3) a nutritional intervention or observational study concerning changes in food and/or beverage intake, including alcohol and/or other nutrition-related behaviors; (4) a comparator of baseline measurements or an alternative diet condition (if interventional); and (5) the outcomes being measurements of dietary and nutrition-related behaviors, disease-related outcomes (e.g., biochemical markers) and/or self-reported symptoms (both interventional and observational designs). To be as comprehensive as possible, all N-of-1 study designs including trials, formative (feasibility and pilot) trials, case studies, and observational studies that were published from 2000 to 2022 were included in the review. The language of publication was limited to those in English. The exclusion criteria were failure to meet the inclusion criteria or insufficient details of study design as an N-of-1 study to ascertain eligibility.

### 2.2. Information Sources and Search Strategy

A systematic search was conducted to identify potentially relevant literature in the following electronic databases: Medline, Embase, Scopus, Cochrane Central, and Psychinfo, from January 2000 until April 2022. Combinations of keywords (N-of-1, single-patient, diet, nutrition, consumption) and their synonyms were searched using appropriate Boolean Operators and advanced search syntax. The Appendix A shows the search strategies used for each of the databases. Additionally, to ensure maximal consideration of the current literature, the reference lists of the articles retrieved and existing narrative reviews were examined for relevant articles, and the contributions of authors known to publish in the area were searched.

### 2.3. Selection Process

Records identified from the literature search were downloaded into EndNote v.20 (2022 Clarivate) and imported into Covidence software (the screening and data extraction tool developed by Cochrane researchers in Australia; Veritas Health Innovation, Melbourne, Australia) for the removal of duplicates, screening, and the full text publication selection process. Initially, titles and abstracts were screened to determine eligibility based on the inclusion/exclusion criteria. Thereafter, the full texts of apparently suitable studies were retrieved and read to establish if they should be included. Both screening stages were conducted independently by two investigators (T.V. and B.B.), and a third investigator (M.A.-F.) resolved discrepancies.

### 2.4. Data Collection Process and Data Items

A data extraction form was developed based on the Cochrane Handbook for Systematic Reviews of Interventions (Version 6.3) [21]. The form was trialed and modified as required to ensure the extraction of all relevant research details. The data items included author(s), publication date, country, study aim, participant demographics and clinical status, N-of-1 methodology, dietary intervention(s), comparator(s), and outcome(s).

### 2.5. Assessment of Quality

Tools were selected to assess the quality of reporting of the studies overall and for the dietary assessment methods employed. For trials and pilot/feasibility studies, the CONSORT Extension for N-of-1 Trials (CENT) 2015 checklist was used [22]. For the observational N-of-1 studies, the STrengthening the Reporting of OBservational Studies in Epidemiology guidelines (STROBE) were selected [23]. The 44 sub-items in the CENT checklist and the 22 items in the STROBE statement were included as data items in the data extraction form [22,23]. Checklist items were arranged under the following broad headings for both CENT 2015 and STROBE: Title and abstract; Introduction; Methods; Results; Discussion; and Other information. For each study, a rating was applied under each broad heading based on the aggregate responses for individual items as follows: yes (addressed all the heading items); most (reported at least half but not all the heading items); few (addressed less than half the heading items); no (reported none of the heading items); or not applicable. All data items were independently populated for each study by two reviewers (T.V. and B.B.) and reviewed by a third investigator (M.A.-F.). Any unclear or missing information was assumed not to be collected or reported.

The quality of the dietary assessment methods reporting was evaluated using an adaptation of a previous tool used by Burrows et al. [24] and later modified by Wang et al. [25]. The original tool did not include ecological momentary assessment methods (EMA), and preceded the more widespread use of technology-assisted methods such as automated recalls (see Appendix A for the tool). The tool includes an assessment of the appropriateness of the method selected, and then awards points for the quality of the validation study and statistics and for data collection methods and analysis, including use of appropriate nutrient databases. For the EMA modification, points were awarded for four or more days of recording; consideration of weekend and weekdays; if the list of questions for dietary assessment was supplied; and for reporting compliance with prompts to submit data, i.e., completeness of data collection. For each study, the items of the dietary assessment tool were extracted, and points were assigned for a score out of 6.0. It was determined that a score of ≥5.0 was good quality; ≥3 to <5.0 acceptable quality; and <3 poor quality. Three authors independently assessed the studies using this tool, and any discrepancies were resolved by discussion.

### 2.6. Data Synthesis

A narrative review was used to synthesize findings related to the included N-of-1 studies’ characteristics, interventions, and outcomes, as well as the type and quality of their selected dietary assessment method and the overall quality of reporting. A commentary was provided to further progress the use of this study design.

## 3. Results

### 3.1. Study Selection

Figure 1 shows the study selection process for this systematic review [20]. From the database searches, 277 potentially eligible records were retrieved, and an additional 9 articles were identified from reference lists and the search of authors known to publish in this research area. After removing 75 duplicates, 202 records were screened based on title and abstract, and thereafter, the full text of 41 records was assessed for eligibility, and of these, 3 were deemed to meet eligibility criteria. Of the 9 additional articles from searching reference lists and authors, 4 were included as suitable, meaning 7 studies were selected for inclusion in this systematic review.

### 3.2. Characteristics of Selected Studies

Table 1 summarizes the characteristics of the selected studies. Three studies were conducted in the USA [26,27,28] and one each in Greece [29], China [30,31], The Netherlands [32], and Scotland [33].

Across the studies, participant numbers ranged from 1 to 28. Ages were reported as ranges or precise age in years. The youngest age range was 16 to 20 years, and the oldest 66 to 70 years. Our age inclusion criterion was studies in adults (i.e., defined as ≥18 years), but for Kwasnika et al., as only one of 25 participants was aged in the 16 to 20 years group, we decided to include this paper [33]. Three studies included males only [26,29,32], and of the studies that included both genders, all had females as the majority [27,28,30,33]. Four studies reported the baseline BMI of participants, which ranged from 17.2–44 kg/m^2^ [28,29,30,32]. Two studies included apparently healthy participants [26,30], whilst subjects in the other studies had a variety of health conditions, including pre-diabetes or Type 2 diabetes [28,29], Hemophilia A, Hemophilia B or Factor VII Deficiency [32], food intolerance [27] and history of alcohol dependence [33].

Three studies aimed to assess individualized responses to dietary interventions [26,29,30]. Three studies assessed the feasibility of various experiments including self-experimentation for irritable bowel syndrome food triggers, the use of EMA to measure the goal of achieving substitution of high glycemic index (GI) foods with low GI foods, and shared medical appointments for healthy lifestyle advice for overweight people with hemophilia [27,28,32]. The observational study examined the social, economic, physical environment, mood, stress, and behavioral factors and policies that affect an individual’s consumption of alcohol [33].

Five studies employed an N-of-1 trial design (with study durations ranging from 72 days to 243 days) [26,28,29,30,32]. One study utilized an N-of-1 observational study design (with an average duration of participation of 64 (SD = 52) days) [33], and one study was a feasibility study of a smartphone app which utilized an N-of-1 trial design (of which the study duration was 12 days) [27].

Interventions included a precision diet for pre-diabetes/Type 2 diabetes [29], isocaloric diets consisting of varying fat and carbohydrate proportions [30], isocaloric low-carbohydrate, low-fat, and very-low-fat vegan diets [26], weekly two-hour shared medical appointments for nutrition and physical activity advice for patients with hemophilia and high BMI [32], potential irritable bowel syndrome food triggers [27], and group-based education followed by individual counselling for goal attainment for low-GI food consumption [28]. The observational study assessed the impact of the implementation of alcohol minimum unit pricing on alcohol consumption as its primary aim [33].

Three of seven studies assessed food and/or beverage consumption as a primary outcome [28,32,33]. One study supplied all foods for consumption with a predefined nutrient composition to test metabolic impacts [30], and one single-case N-of-1 trial documented typical menus for three diets [26]. Three studies observed body weight change [26,29,32], and two measured blood glucose concentrations [29,30], one of which measured additional biochemical and microbiome measurements [30]. One study examined blood pressure [29] and one observed anthropometric change (body shape, waist circumference) [26]. Four studies assessed general physical, psychological, and social factors (i.e., physical activity, motivation, stress, and social contact) [26,27,32,33]. Three studies had feasibility/usability and compliance measures in their outcomes [27,28,32].

### 3.3. Quality Assessment of Reporting and Dietary Assessment Methods of the Selected N-of-1 Studies

Table 2a shows the results for the quality assessment of reporting for the N-of-1 trials included in this systematic review, based on the CENT 2015 checklist. The results for assessment of the observational N-of-1 study (based on STROBE) are included in Table 2b. With regard to the title and abstract items, five studies reported all items, one study reported most items [28], and one study reported no items [26]. All studies reported all introduction-related items. For method items, one study reported all items [32], five reported most items and one study reported less than half the items (few) [26]. For results, five studies included most items, one study reported all [33] and another only a few items [26]. All discussion-related items were reported by all studies, while further information was only fully reported by three studies [29,30,33], with two others reporting most items [27,32] and two reporting few items [26,28].

It is worth noting certain items were poorly reported by multiple studies with respect to methods and results. All but two studies failed to report the dates defining their periods of recruitment and follow-up [30,33]. Only two of the six trials mentioned whether unintended harms were considered or measured [29,32].

Four dietary assessment methods were used across the studies: two were 24 h recalls [28,29]; one was a single diet question administered three times weekly (“How healthy have you eaten in the past two days?”) [32]; one had all food of known composition provided and a checklist for compliance [30]; and three were EMA dietary protocols [27,28,33]. Five studies used one method each, one article employed two methods [28] and one study did not include an assessment method at all [26] (see Table 3).

None of the studies reported a validation method/reference for their dietary assessment, but for one study, the use of standard recipes and dietitian-constructed menus with a full analysis of nutrients using an appropriate database and the fact that all foods were supplied meant that it did not necessitate traditional validation [30]. EMA methods were self-administered [27,28,33], the single dietary question was self-administered [32], and 24 h recalls were conducted by trained researchers [28,29]. Three studies reported they had trained dietitians involved in dietary analysis [28,29,30]. Based on the modified tool, the overall dietary assessment quality rating derived from the methods was poor to good, as can be seen in Table 3.

## 4. Discussion

Our literature search revealed only a small number of nutrition-related studies have used the N-of-1 study design, and half the studies were feasibility or pilot studies. Thus, the field appears to be in its infancy for medical nutrition therapy and nutrition promotion. A total of 58 people were studied in the trials, which included healthy young people, people with prediabetes and Type 2 diabetes, people with irritable bowel syndrome, and people with hemophilia who were suffering from overweight/obesity. A further 25 people were investigated in an observational study concerning alcohol behaviors. While the study number is small, this collection of studies provides a starting point to discuss future possibilities and suggestions to improve the conduct and reporting of N-of-1 trials to increase their acceptance as an appropriate way to test medical nutrition therapies and advice for healthy eating in nutrition promotion. This includes a discussion of the gold standard N-of-1 design (a double-blind randomized trial); the observational N-of-1 study design; the application of genomics for personalized/precision nutrition; engagement of patients/individuals to be active participants in their trials; technologies that will assist N-of-1 studies that were not possible when Guyatt et al. [9] first proposed N-of-1 designs; and the growing evidence base that will assist the field of nutrition to be conscious of psychosocial and environmental determinants and biological developments in genomics and microbiota.

The largest trial applied the N-of-1 randomized double-blind cross-over trial design with a baseline period and washout periods (six days) before cross-over to the alternate diet, with three cycles of the dietary regimes (known as AB BA design × 3) [30]. The trial serves as an example of the application of the N-of-1 design to study both the individual and group effects of different diet compositions on metabolic responses under strictly controlled conditions with all meals provided. Testing the metabolic response to diets in this short-term way can guide follow-up personalized nutrition advice that can be assessed in a trial of a longer duration. Gkouskou and colleagues have completed an N-of-1 design to test genetically guided nutrition therapy versus conventional medical nutrition therapy in a quasi-experimental AB design (conventional followed by genetically devised) with impressive outcomes in an eight week period, including reversal of prediabetes [29]. This was in contrast to the large Food4Me RCT [34], which was a European four-arm RCT in 1600 participants across seven countries, testing three levels of personalized advice: (1) that based on current diet; (2) that based on current diet and phenotype data, e.g., serum cholesterol concentration; and (3) that based on current diet, phenotype and genotype data. These were all compared with standard healthy eating following dietary guidelines as the control. All three forms of personalized advice led to improvements in nutrient composition and overall healthiness of the diet after six months, but none were demonstrated to have an advantage over the others [34]. However, as the senior author commented in a follow-up publication, the very nature of the RCT reduces the analysis to a comparison of mean changes, and he suggested there is merit in N-of-1 studies to further understand personalization of diet [35]. The consensus of the American Academy of Nutrition and Dietetics on genetic testing in medical nutrition therapy has concluded that the current evidence base is limited and weak, based on their systematic review which was inclusive of 12 RCT studies [36]. One suggestion is that practitioners collect and store data on their Health Informatics and Infrastructure site (ANDHII) in an ‘N-of-1’ fashion for later synthesis and evaluation. The success of the Gkouskou et al. study [29], albeit in three patients, should impart researchers and dietitians with some confidence to perform similar studies to expand the range of diseases and evidence base. Certainly, data spanning patients with different demographic, environmental and psychosocial attributes as well as biological factors will further the study of personalized nutrition. The N-of-1 study design obviously allows for advances in our understanding of individuals responses to any given medical nutrition therapy.

Nutrition science by its very nature as a science may fail to consider many contextual factors concerning the social, emotional, and food and physical environments that dynamically influence food consumption. It is possible to change one’s food behaviors, but almost impossible to change genetic biology. A more comprehensive approach to studying the multiplicity of factors contributing to an individual’s response to food and beverage consumption is indicated [35,37]. The N-of-1 study design provides this opportunity. An individual’s food and health literacy and ability to self-regulate all contribute to the maintenance of a prescribed dietary pattern, and these factors are not usually measured in metabolic studies of nutrient responses, even when conducted for extended periods [38,39]. Dietitians are trained to consider as many of these factors as possible in practicing medical nutrition therapy, but they would gain an advantage by having a greater insight into the behaviors of their patients over longer durations [19]. An ability to focus in on periods outside dietetic consultations in real time, to study predictors and patterns of food consumption and the dynamics of consumption that may be in a steady state or one of continual flux, might prove useful [37]. It is almost impossible for patients to recall and self-report the many daily changes in conditions they face when making decisions about food, but technology offers the ability to take multiple measurements of numerous factors (some of them passively, without burden to the individual), albeit food and beverage consumption itself can be challenging and will be further discussed below [37].

The observational study of Kwasnika et al. serves as an example of how contextual factors can be studied repeatedly over time using the N-of-1 study design to gain a more complete picture of all factors influencing consumption behavior, although in this case, it is alcohol [33]. The rich data set that can be collected on an individual with continuous sampling over time enables fine-grained tailoring of plans and counselling for patients, and could potentially empower individuals to gain greater insight into their own diet-related behaviors for purposes of self-regulation and building self-efficacy to deal with challenges as they arise [40]. A patient may have a better metabolic (biological) response to a diet with a given ratio of macronutrients, but prescription of an optimal diet must pay heed to the reality of the multiplicity of factors that makes following a given diet achievable. Studying these factors and responses over time, for example, in an N-of-1 observational study, may allow the patient to select an acceptable compromise between the ‘ideal’ diet and that which can be not only tolerated but also enjoyed long term. Health behaviors may explain 40% of the variance in health outcomes, compared with the 30% contribution of our own biology (with social determinants making up the other 30%) [8]. There is the possibility of designing studies that allow continuous fine-tuning of the intervention as more is learned about the individual with continual data capture [8,41].

One of the other stated advantages of N-of-1 studies is that individuals can be active participants in their treatment. While randomization to drugs and placebo with blinding may be best for testing pharmaceuticals, in the real world, people mostly cannot be blinded to the food they eat. N-of-1 randomized trials with blinding are possible for testing acute and short-term effects of foods and meals provided from a metabolic kitchen setting, but not for testing long term medical nutrition therapy. Although bias may be introduced, the opportunity for individuals to select diets and conduct their own N-of-1 experiments may be feasible, as shown by the study of Feltham and Westman [26]. Karkar et al. have demonstrated how smart technology might be used by patients to experiment and test individual dietary triggers for irritable bowel disease [27]. Kravitz et al. demonstrated it is possible to recruit a large (if somewhat biased) population (n = 447) to conduct their own cross-over N-of-1 trials to test treatments to enhance cognitive and emotional wellbeing [42]. Interestingly, those participants with the lowest expectations of the treatment experienced the greatest gains [42].

In the past decade, developments in technology have provided infrastructure, remote sensing, and communication to support N-of-1 studies by clinicians, researchers, and individuals themselves. Yet, the uptake of N-of-1 designs in nutrition research appears very limited. Remote technologies to validly assess diet continue to evolve [43,44]. Behavioral measures of how people eat, such as frequency of eating and length of an eating episode, may be gleaned using wrist accelerometry [45], and the number of bites and eating speed may be measured using an app and smartwatch [46]. Direct measures of the biochemical and physiological outcomes of medical nutrition therapy, such as blood glucose concentrations (as demonstrated in the Ma et al. study [30]), blood pressure, and body weight [14], can be conducted remotely, passively, and continuously. Dietary intake information alongside factors such as food environment, and social, and emotional contexts can all be measured simultaneously using smart technology [14]. The method of EMA that prompts the participant at regular intervals throughout the day to share consumption information via data text entry or uploading of digital food images has been validated in several studies [37,47].

The main deficiency in the dietary assessment of the studies included in this systematic review was a lack of validation of their tool, even when a comparison of two methods was available. For example, in the study by Miller et al. comparing EMA assessment of low-GI foods and 24 h recall would have allowed for relative validity to be determined using statistical methods such as correlation and Bland–Altman; however, this does not appear to have been conducted [28]. When designing dietary assessment for an N-of-1 study, the purpose should be defined and the method selected accordingly. There is a balance between burden and granularity of data and the aims of the study, and the motivation to assess food and beverage consumption should guide the design of the assessment [37]. The motive of the study of Miller et al. was to determine the success of patients with Type 2 diabetes mellitus in achieving their goals to replace high-GI foods with foods low in GI [28]. To track progress, there was no need to ask patients to report their total food intake every day. Rather, EMA three times daily was a useful way to regularly collect data on the low-GI foods, and a whole-of-diet approach was measured with 24 h recalls at baseline and the end of each of the two six-week study phases; only the relative validation detail was omitted [28]. While the excessive burden of recording total diet for prolonged durations is discouraged, equally, too little data collection may result in failure to capture meaningful changes in food consumption. To minimize participant burden, Hendrik et al. asked patients only a single question every two days concerning their perceived healthiness of their eating via a response on a visual analogue scale [32]. However, if additionally, more detailed questions were asked on two or three occasions, such as short questions about fruits, vegetables, wholegrains, and ultra-processed food consumption (that exemplify healthy versus unhealthy foods), they may have detected differences. For medical nutrition therapy, the aim is frequently for substitution or modification of specific foods; thus, the measurement of food and diet in N-of-1 studies should fit this intent, and should be considered in that light when assessing the quality of assessment. On some occasions, total daily nutrient intake is clearly required. For instance, the percentage of energy from protein and other macronutrients is needed to study protein leverage, so a more comprehensive assessment with a traditional method is indicated [48]. Perhaps further nutrition-specific guidelines could be developed for N-of-1 studies and added to the existing reporting guidelines of CENT [22]. Some suggested recommendations around reporting of EMA as a dietary assessment method were published in 2016, and could be built upon for N-of-1 studies when using this method for data collection [49].

The studies reviewed demonstrate the methods of data analysis that can be employed in N-of-1 studies. For individuals, the data from numerous measurements can be depicted in graphical form, and inspection of plots and summary estimates from each period will reveal the better diet option [27,28,32,33]. Due consideration to order and time effect are needed. When aggregating data from multiple patients to understand group effects within a trial, or for a meta-analysis of similar trials, the two approaches most commonly employed are linear mixed models and a hierarchical Bayesian model [50].

The strengths of the current review include the extensive literature search of five databases, and the searching of reference lists of publications retrieved, prior reviews and opinion pieces, and authors identified as publishing in the area; the review was as inclusive as possible. We also undertook a quality assessment of the studies’ reporting and the dietary assessments. The limitations of this review are that only studies published in English were included, and the study participants were restricted to adults. In young children, the reporting of nutrition behaviors and contextual factors becomes the responsibility of parents/carers, and real time capture of data, for example, using an EMA approach, would be difficult. A clinical service for N-of-1 studies in children using complementary and alternative medicine (CAM) has existed in Canada for almost two decades, but CAM is not medical nutrition therapy [51]. The authors also acknowledge that the current attempt at selecting criteria for optimal dietary assessment by EMA in the existing tool was a first attempt, and validation of dietary assessment in EMA is an area for further investigation and consensus building [47]. The small number of participants and heterogeneity of design, aims and outcomes among studies did not offer the opportunity for any data aggregation for meta-analysis. With respect to the studies included, the overall quality of reporting of studies according to CENT 2015 showed the greatest deficit was failure to describe methods adequately. Registering a study or publishing a protocol would help to clarify some of the forgotten items, and this was an area neglected by some authors. Description of the study methods and the reporting of results are important for making decisions about the internal validity of a study.

The future will reveal if N-of-1 studies show increased adoption in testing and informing medical nutrition therapy and preventive nutrition. We will likely see researcher investigations into testing of metabolic and microbiota responses and personalizing diets to manage long-term chronic diseases. The comprehensive big data from the Precision Nutrition initiative in the US should expand the evidence base around how nutrients, foods, and dietary patterns predict favorable profiles of microbiota and gene-food/nutrient interactions to solve some of the deficiencies in current understanding. The big data can be used to inform choices in personalized N-of-1 studies, and conversely aggregated knowledge gained from N-of-1 studies used to further data mining from the million in the precision initiative [52]. The public is interested in self-monitoring their health, and self-experimentation with N-of-1 design is possible. As wearables, devices, and sensors become more reliable and costs decrease, greater usage will be expected. There is an opportunity for dietitians working in clinics and private practices to use the N-of-1 method and contribute to research; indeed, many would attest they already engage in this type of practice, but not with systematic documentation of the process. Several centers established to support drug trials among medical practitioners recognized the importance of resourcing and providing infrastructure, as managing N-of-1 studies was considered burdensome in general practice [53]. Supportive resources and a register may be required for dietitians to employ N-of-1 studies for testing medical nutrition therapy.

## 5. Conclusions

While limited studies have been published to date, the N-of-1 study design is a potential way forward for dietitians to test holistic personalized medical nutrition therapy informed by precision evidence, and to further the study of the dynamics of eating and the many personal and contextual influences that shape what and how an individual eats and drinks. Medical nutrition therapy, because of its grounding in both the biological and behavioral sciences, may have more to gain than pharmaceutical studies did. Using an observational design to study behaviors before planning interventions based on biology would appear useful to progress precision nutrition to create true personalization. Such an approach will advance optimal nutrition education and counselling that respects the uniqueness of each person and is necessary for individuals to reach their dietary goals.

## Figures and Tables

**Figure 1 nutrients-15-01756-f001:**
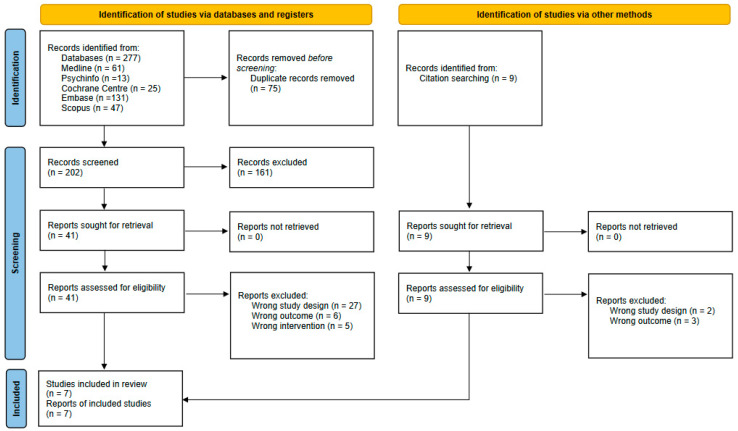
PRISMA flow diagram for searches of databases, registers and other sources for N-of-1 study designs, from [20].

**Table 1 nutrients-15-01756-t001:** Characteristics and outcomes of the selected studies.

Authors, Year, Country	N	Participant Characteristics	Aim/s	Study Design	Intervention/s	Comparator/s	Outcome/s
**Interventional Studies**							
Feltham and Westman, 2021, USA [26]	N = 1 M	Age: 29 y	To determine any differential effects of hypercaloric feeding of 3 diets on body composition	N-of-1 3 × 21 d periods (ABC) separated by 3 mo wash-out Analysis Differences in repeated measures of weight and waist before and after diet	5800 kcal diets Low-carb: 6% CHO 72% fat 22% protein Low-fat: 64%CHO 23% fat 13% protein Very-low-fat vegan: 68% CHO 15.5% fat 16.5% protein	Baseline measurement	Caloric intake (Δ per 21 d): 121,676 kcal (low-carb) 121,653 kcal (low-fat) 121,674 kcal (very-low-fat vegan) Weight Δ kg: 1.3 (low-carb) 7.1 (low-fat) 4.7 (v low-fat vegan) Waist Δ cm: −3 (low-carb) 9.25 (low-fat) 7.75 (v low-fat vegan)
Gkouskou et al., 2022, Greece [29]	N = 3 M	Age: 45, 54, 67 y BMI: 23.5, 26.2, 31.5 kg/m^2^ Pre-diabetes (n = 2) Type 2 diabetes (n = 1)	To assess the effectiveness of precision genetically guided MNT interventions vs. conventional MNT	N-of-1 quasi-experimental, AB cross-over trial 8 weeks conventional MNT (A) 1 week wash-out 8 weeks precision MNT (B) Analysis Differences in percentage change in measures from baseline to end of each MNT period	Precision diet based on nutrigenetic testing: 1800–2200 kcal/day 25–32 g fiber 45–49%E CHO 26–37%E fat 16–25%E protein 7–9%E SFA	Conventional MNT: 1750–2200 kcal/day 33 g fiber 45–50%E CHO 20–32%E fat 18–30%E protein 7%E SFA	%Δ body weight: I = −1.4 to −7.4, C = 0 to −4.9 %Δ SBP/DBP: I = 0 to −8.6/−1.4 to −5.9, C = 0 to −3.4/0 to −1.2 %Δ HbA1c: I = −3.6 to −8.5%, C = 0% %Δ FPG: I = −6.7 to −25.5, C = 0 to −6.5
Hendriks et al., 2021, Netherlands [32]	N = 5 M	Age: 30–64 y BMI = 29.61–38.88 kg/m^2^ Hemophilia A (n = 3), Hemophilia B (n = 1) Factor VII Deficiency (n = 1)	To evaluate feasibility and efficacy of shared medical appointment in people with hemophilia to improve PA and eating habits (diet)	Randomized A B A’ N-of-1 trial staggered phase Baseline: 2–4 weeks A Intervention: 7 weeks B post-intervention: 2–4 weeks A’ Analysis Visually using 2-SD band method or Randomisation test	Weekly 2-h shared medical appointments using multiple behavioral change techniques	Baseline measurements	Primary: PA increased Diet (1/5 patients) Secondary: pain in rest NS pain during PA NS, patient-specific complaints NS, kinesiophobia NS motivation for PA and diet NS Decrease BMI 2/5 Decrease BP 2/5
Karkar et al., 2017, USA [27]	10 F 5 M	Age: 20–69 y Food intolerances Rome IV IBS criteria	To evaluate the feasibility of TummyTrial app to conduct self-experiments for diagnostic self-tracking	Feasibility study on TummyTrials, an app that enables N-of-1 study design. Alternating Treatment Design: AB at random for 12 d Analysis Visualization of symptoms with timeline plots and trend plots	Breakfast with potential IBS food trigger (B)	Breakfast without potential IBS food trigger (A)	Participant compliance: 12/15 100% compliance for the 12 d Change in Pre- and post-IBS Symptom severity score NS Feasible and acceptable to patients
Ma et al., 2021, China [30] Tian et al. [31]	19 F 9 M	Age: 22–34 y BMI: 17.2–31.9 kg/m^2^ Healthy	To investigate individual variability in postprandial glycemic response to diets with different proportions of Fat:CHO	N-of-1 randomized trial 6 d washout Then, three cycles of 3 d treatment 6 d washout 3 d treatment AB BA AB BA AB BA Analysis Difference in means between diets for individuals. Determined clinical meaningful difference. Further analysis of group effects using a hierarchical Bayesian model	Isocaloric diets HF-LC: Fat 60–70%E CHO 15–25%E Protein 15%E LF-HC: Fat 10–20%E CHO 65–75% Protein 15%E	Baseline measurements of individuals	Primary: MPG 10/28 participants showed clinically meaningful difference (3 HF and 7 HC responders) MAGE 9/28 clinical difference (4 HF and 5 HC responders) AUC24 NS Secondary: collected microbiome and urine metabolomic profiles (NR)
Miller et al., 2016, USA [28]	4 F 2 M	Age: 58–62 y BMI: 25–44 kg/m^2^ T2DM ≥ 1 y no insulin therapy	To assess the feasibility of using mobile EMA to monitor low GI food intake and goal attainment after group education and individual counselling	N-of-1 pilot study AB 6-week group education A 6-week goal setting and individual counselling with EMA for data collection and monitoring B Analysis Changes in servings of low GI foods	Group-based education followed by individual counseling for goal attainment to increase low GI foods	Serves of low GI foods at baseline	Compliance: 79.3% EMA prompts completed; 16.4% ignored; 4.2% refused Servings of low GI foods: Mean Increase (SD) 1.2 (0.1) GI goal attainment 3/6 participants Acceptable and feasible program
**Observational Study**							
Kwasnicka et al., 2020, Scotland [33]	16 M 8 F 1 NI	Age: 16–65 y Current or recent history of alcohol dependence	To assess interpersonal differences in psychological and social factors associated with daily alcohol intake and effects of maximum unit price (MUP) and contextual factors	N-of-1 observational study 3 × 12-week waves of data collection pre-MUP (n = 11) pre- and post-MUP (n = 11) post-MUP (n = 3) used mixed methods as interviews after n-of-1 observation 15/25 analyzed Analysis Visualisation of changes over time by individuals. Regression and multilevel models for combined data	NA observational	NA observational	Amount and type of alcohol consumed: small decrease after MUP implementation amongst heaviest drinkers Less difference amongst less frequent or mainly abstinent drinkers Variation among participants as to internal and external factors associated with alcohol intake

Footnotes: AUC24 refers to the total area under the continuous glucose monitor curve from 00:00 to 24:00 of the day; CHO, carbohydrate; △, change; C, comparator; DBP, diastolic blood pressure; EMA, ecological momentary assessment; %E, percentage of total energy; F, female; FPG, fasting plasma glucose; GI, glycemic index; HC, high carbohydrate; HF, high fat; IBS, irritable bowel syndrome; I, intervention; M, male; MPG, maximum postprandial glucose; MAGE, mean amplitude of glycemic excursions; MNT, medical nutrition therapy; MPG is the peak value of CGM within 3 h after the first bite of a meal or the maximum CGM value between 2 meals when the interval is <3 h; NI, not identified; NR, not reported; NS, no significance; PA, physical activity; SBP, systolic blood pressure; SFA, saturated fatty acids; T2DM, Type 2 diabetes mellitus; wk, week.

**Table 2 nutrients-15-01756-t002:** Quality assessment of reporting of items in the CENT 2015 checklist [22] or STROBE checklist [23].

Author, Year Reference	Title and Abstract	Introduction	Methods	Results	Discussion	Other Information
(a) N-of-1 trials (including feasibility and pilot studies)						
Feltham and Westman 2021 [26]	No	Yes	Few ^a^	Few ^a^	Yes	Few ^a^
Gkouskou et al., 2022 [29]	Yes	Yes	Most ^b^	Most ^b^	Yes	Yes
Hendriks et al., 2021 [32]	Yes	Yes	Yes	Most ^b^	Yes	Most ^b^
Karkar et al., 2017 [27]	Yes	Yes	Most ^b^	Most ^b^	Yes	Most ^b^
Ma et al., 2021 [30] Tian et al., [31]	Yes	Yes	Most ^b^	Most ^b^	Yes	Yes
Miller et al., 2016 [28]	Most ^b^	Yes	Most ^b^	Most ^b^	Yes	Few ^a^
(b) N-of-1 observational study						
Kwasnicka et al., 2020 [33]	Yes	Yes	Most ^b^	Yes	Yes	Yes

^a^ Few means that less than half of the recommended items were included. ^b^ Most means more than half of the recommended items, but not all items, were included.

**Table 3 nutrients-15-01756-t003:** Quality rating of reporting of dietary assessment method(s) employed.

Author, Year	Dietary Assessment Method	Validation	Data Collection	Data Analysis	Score	Overall Rating
Feltham and Westman, 2021 [26]	NR	NR	Participant used nutrition information on packaging and a supermarket website to determine calorie and macronutrient compositions	NR	0	Poor
Gkouskou et al., 2022 [29]	24-h Recall	NR	Administered by a trained researcher via phone interview, but did not specify whether subjects were trained for data collection. The 24 h recall was reviewed/checked by a trained person. Did not specify days of recall, nutrient database(s) used nor comment on aids or multiple passes	Data coded and analyzed by a trained individual	1	Poor
Hendriks et al., 2021 [32]	Single Diet question on a visual analogue scale	NR	Self-reported eating habits via visual analogue scale were completed digitally in a secured data entry platform. Did not specify whether subjects were trained for data collection. Dietary Questions provided.	Unclear, but appears principal researcher carried out the analysis	1.25	Poor
Karkar et al., 2017 [27]	EMA questions	NR	Self-reported compliance with experimental menu condition via TummyTrials app), subjects were trained for data collection. Prompts to complete and compliance assessed by app. Weekend and weekdays were considered at six days’ recording. Dietary questionnaire (app EMA) supplied within text	Analysis built into app system and documented by researchers (consultation with dietitians)	3.0	Acceptable
Kwasnicka et al., 2020 [33]	EMA questions	NR	Self-reported alcohol consumption via smartphone survey; subjects were trained for data collection and the data reviewed by researchers as appropriate for this type of survey. Daily surveys sent at 7 p.m. to the mobile phones of participants or a study phone for 12 weeks, weekend and weekdays considered, and authors reported compliance with EMA prompts. Questions on amounts and types of alcohol in the previous 24 h. Questionnaire (EMA) supplied.	The researchers conducted the analysis	3.0	Acceptable
Ma et al., 2021 [30] Tian et al., 2020 [31]	Weighed food (known composition) provided and food checklist to track	Weighed food provided for which nutrient composition calculated and standard recipes used	All food provided via the University canteen. Diets designed by dietitian and participants sign in to show they consume the meal provided. Food checklist not provided and did not discuss method of checking; however, compliance with meals reported as 98%	Dietitian prepared diet of known composition. Did not specify whether checklist coded and analyzed by a trained individual	5.5	Good
Miller et al. 2016 [28]	24-Hour Recall	NR Unclear if validated by University of Minnesota service	Administered by trained staff via an interactive phone interview and collected unannounced. Nutrient database reported. Multiple days of recall (1 weekend day and 2 weekdays selected at random), multiple pass approach used, and a food amounts booklet was provided to help participant estimate portion sizes.	Analysis and coded by the University of Minnesota 24 h recall service	3.0	Acceptable
	EMA questions	Checklist used to prompt completion. Statistics on agreement with 24 h recall days not reported	Self-reported servings of low GI foods via PRO-Diary EMA; subjects were trained for data collection. Checklist of low-GI foods used to assist entry and memory. 6 weeks of EMA recording, weekend and weekdays considered, and authors reported compliance with EMA prompts	Did not specify whether coded and analyzed by a trained individual	3.0	Acceptable

NR: Not reported.

## Data Availability

No new data were created in this study. Data sharing is not applicable to this review.

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
