# Peer review of "Nutrition-Related N-of-1 Studies Warrant Further Research to Provide Evidence for Dietitians to Practice Personalized (Precision) Medical Nutrition Therapy: A Systematic Review"

_nutrients, 2023, doi:10.3390/nu15071756_

Round 1

Reviewer 1 Report

Dear Authors,

I congratulate you on the idea and appreciate the work you put into preparing the review and report of the available case studies. Previous population or group nutritional studies provide averaged results that do not show any particular individual differences. You understand this perfectly and I share your view on developing the gold standard of N-of-1 nutritional research.

I have no objections to the way the manuscript was prepared.

Best regards

Author Response

Dear Reviewer

thank you very much for taking the time to review our paper and the comments.

Reviewer 2 Report

This manuscript systematically reviews nutrition-related studies using an N-of-1 design with study qualities assessed. The evidence base is small but illustrates the main characteristics of N-of-1 study designs with discussions about the use of such study design to move research forward. The review follows a good organization and flow, and the aims are addressed. I only have some minor questions for the authors to address before consideration for publication.

1. For the eligibility criteria, sample size was not mentioned, and one study with only n=1 was included in the review. Why not consider sample size as one of the inclusion criteria? Does a power analysis required if the study follows the N-of-1 study design? Would the results be representative and generalizable if there is only one subject recruited?

2. The authors mentioned somewhere in the discussion that a meta analysis was not performed. Would it be possible to generate a meta analysis as some studies measured same outcomes and observed clinical changes?

3. In the discussion section the authors mentioned one of the marked advantage of the N-of-1 study design. What are other advantages? It would be better to discuss more regarding the advantages and practical implications of this study design. 

Author Response

Thank you very much for reviewing our manuscript.

We have reviewed each of your three questions and made track changes in the manuscript to address these as outlined in the attached PDF.
